# Leaf functional traits and pathogens: Linking coffee leaf rust with intraspecific trait variation in diversified agroecosystems

Stephanie Gagliardi[1], Jacques Avelino[2,3], Adam R. Martin[1], Marc Cadotte[4], Elias de Melo Virginio Filho[5], Marney E. Isaac[1] *

1 Department of Physical and Environmental Sciences, University of Toronto Scarborough, Toronto, Ontario, Canada, 2 CIRAD, UMR PHIM, Montpellier, France, 3 Institute Agro, PHIM, University Montpellier, CIRAD, INRAE, IRD, Montpellier, France, 4 Department of Biological Sciences, University of Toronto Scarborough, Toronto, Ontario, Canada, 5 Centro Agronómico Tropical de Investigación y Enseñanza, Turrialba, Costa Rica

* marney.isaac@utoronto.ca

## Abstract

Research has demonstrated that intraspecific functional trait variation underpins plant responses to environmental variability. However, few studies have evaluated how trait variation shifts in response to plant pathogens, even though pathogens are a major driver of plant demography and diversity, and despite evidence of plants expressing distinct strategies in response to pathogen pressures. Understanding trait-pathogen relationships can provide a more realistic understanding of global patterns of functional trait variation. We examined leaf intraspecific trait variability (ITV) in response to foliar disease severity, using *Coffea arabica* cv. Caturra as a model species. We quantified coffee leaf rust (CLR) severity—a fungal disease prominent in coffee systems—and measured key coffee leaf functional traits under contrasting, but widespread, management conditions in an agroforestry system. We found that coffee plants express significant ITV, which is largely related to shade tree treatment and leaf position within coffee canopy strata. Yet within a single plant canopy stratum, CLR severity increased with increasing resource conserving trait values. However, coffee leaves with visible signs of disease expressed overall greater resource acquiring trait values, as compared to plants without visible signs of disease. We provide among the first evidence that leaf traits are correlated with foliar disease severity in coffee, and that functional trait relationships and syndromes shift in response to increased disease prevalence in this plant-pathogen system. In doing so, we address a vital gap in our understanding of global patterns of functional trait variation and highlight the need to further explore the potential role of pathogens within established global trait relationships and spectra.

**Data Availability Statement:** The data underlying the results presented in the study are deposited in the Dryad Digital Repository https://doi.org/10.

## Introduction

Over the last few decades, there have been a growing number of studies quantifying and describing plant functional trait relationships among and within plant species, e.g. [1,2]. Trait

5061/dryad.2bvq83bvp. Trait data used in this study has also been contributed to the TRY Functional Trait Database at try-db.org (DatasetID 722).

**Funding:** This work was supported by the Natural Sciences and Engineering Research Council of Canada (Discovery Grant to M.E.I. and Alexander Graham Bell Canada Graduate Scholarship to S.G.). The funders had no role in study design, data collection and analysis, decision to publish, or preparation of the manuscript.

**Competing interests:** The authors have declared that no competing interests exist.

relationships and trait spectra, such as those comprising the "Leaf Economics Spectrum" (LES) [1], are now commonly interpreted to reflect differences in plant resource-use strategies, such that leaves and plant species differ from one another across a gradient of resource conservation and resource acquisition. Drivers of trait variation include abiotic conditions, biotic interactions, and anthropogenic influences, all of which individually or cumulatively provide insights into the determinants of species distributions [3,4], and enhance our understanding of the role plant diversity plays in governing rates of ecosystem functioning [1,5,6].

However, research on functional trait relationships—and indeed much of the world's functional trait data—has largely focused only on quantifying traits in "healthy" plants or unstressed ecosystems [7]. In fact, standardized methodologies for measuring plant functional traits explicitly call for the exclusion of plants with visible signs of herbivore damage and pathogen incidence [8,9], creating a major gap in global plant functional trait databases. This is despite the fact that plant pests and pathogens are among the most prominent abiotic and biotic pressures on plant communities [10].

Across plant species, key leaf functional traits have been established as proxies for anti-herbivore defensive strategies, including decreased specific leaf area (SLA) and increased leaf dry matter content (LDMC), which negatively impact the palatability of leaves [11]. However, correlations between leaf functional traits and plant pathogens have not been as well established, despite emerging research linking the two and the co-evolution of many plant-pathogen systems. For example, within a single plant species, leaves that express greater SLA are more susceptible to foliar biotrophic fungi (i.e., pathogenic fungi that require living host tissues to survive) [12]. Similarly, intraspecific variability in leaf nutrient concentrations can dictate fungal foliar disease severity [13], such that leaves with greater leaf nitrogen concentrations (LN) are associated with greater foliar biotrophic fungal disease severity [14]. Although these preliminary findings are specific to plant genotypes [15] and pathogen types [16], they indicate that leaf functional trait syndromes can shift in relation to foliar pathogens: more resource acquisitive traits relate to greater susceptibility to and lower tolerance of biotrophic fungal pathogens.

By extension, therefore, we also expect a shift in functional trait relationships towards acquisitive traits in leaves with visible signs of foliar disease, compared to visibly "healthy" leaves. However, since diseased leaves among and between species have been purposely excluded from many of the most prominent analyses on global leaf trait spectra [5], trait-based responses to or interactions with pathogens remain poorly assessed. As a result, we have a vital gap in our current global plant functional trait databases, and an incomplete understanding of how foliar diseases interact with leaf trait expression as well as with established global leaf trait relationships and spectra.

Our study aims to quantify intraspecific trait variation (ITV) in leaves with visible signs of foliar biotrophic fungal pathogen incidence and determine if leaf traits and trait trade-offs are correlated with foliar disease across multiple scales. To do so, we measure seven key leaf traits of a single plant cultivar, across three plant canopy strata, over a nested sampling design comprised of multiple management practices (including shade tree diversity, amendment treatments and yield classes); a sampling scheme that was designed to maximize the extent of leaf trait variability. We use bivariate and multivariate analyses, as well as a novel application of hierarchical clustering analyses to infer relationships between pathogens and ITV. Finally, we compare trait relationships measured within diseased leaves with published trait relationships measured on visibly disease-free leaves of the same genotype and from the same region. We hypothesize that (1) leaves with visible signs of foliar disease differ from one another, from resource conserving through to resource acquiring trait values, among contrasting management practices and location within the plant; (2) resource conserving traits correlate with

reduced foliar disease severity; and (3) trait relationships differ in plants with foliar disease, compared to leaves with no visible signs of foliar disease.

## Model species

*Coffea arabica* cv. Caturra is an internationally significant coffee cultivar that often expresses substantial leaf ITV resulting from both biotic and abiotic factors, including light availability [17], air temperature [18], soil moisture and nutrients [19,20], microbial diversity [21], and reproduction [22]. Additionally, coffee leaf traits have been found to co-vary along an intraspecific LES, with plants showing trait values reflecting resource conservative through to acquisitive strategies [20,22–24]. However, as is likely true for other wild and cultivated plant species (e.g. [25]), virtually all studies on coffee ITV have explicitly excluded plants with visible signs of pests and pathogens [20].

The Caturra variety is highly susceptible to coffee leaf rust (CLR), a foliar disease caused by *Hemileia vastatrix*, a biotrophic fungus that has co-evolved with *Coffea* spp. The lifecycle *H. vastatrix* has been extensively described and reviewed [26–29]. Similar to coffee leaf ITV, microclimate conditions, which are shaped by agroecosystem design [30–32], largely constrain different lifecycle stages of *H. vastatrix* [32,33]. Generally, *H. vastatrix* uredospores, the only important infectious unit of *H. vastatrix* of coffee, germinate on the underside of coffee leaves and penetrate the leaf via stomata given appropriate temperatures, solar radiation, and moisture levels. After colonizing and maturing within the inter- and intracellular leaf space, fungal sori emerge from the stomata and produce new spores that are dispersed largely via wind, rain, and the movement of farm workers.

The essential role of coffee leaf stomata in the infection cycle suggests that greater stomatal density increases *H. vastatrix* incidence [27]. Simultaneously, increased stomatal density is allometrically related to other leaf traits, including lower SLA and greater lamina thickness [34]. Biochemical responses to *H. vastatrix* invasion within coffee leaves include an increased production of defence-related proteins and hypersensitive cell death, though these responses generally occur too late in genetically susceptible coffee varieties, such as Caturra, to prevent negative consequences of CLR infection [35]. Overall, previous studies have found that high-yielding coffee plants are more susceptible to CLR [36], potentially due to the reallocation of defensive phenolic compounds from leaves to fruits [37]. However, our understanding of the role CLR plays in determining functional trait expression and trait-trait relationships is lacking, despite the role trait expression may play in moderating plant-pathogen dynamics or the role that disease may play in manipulating common trait expression.

## Materials and methods

### Site description and experimental design

This study was conducted at the international coffee agroforestry research trial established by the Centro Agronómico Tropical de Investigación y Enseñanza (CATIE), located in Turrialba, Costa Rica (09˚53´44” N, 83˚40´7” W, 685 m a.s.l., with approval from and in collaboration with CATIE's Trial Coordinator. Additional information regarding the ethical, cultural, and scientific considerations specific to inclusivity in global research is included in the (S1 Checklist). Sampling occurred in July 2017, during the rainy season in this region, which averages 3200 mm of annual rainfall [38]. The CATIE site is organized into three distinct blocks, with different soil amendments and shade tree treatments repeated within each of these blocks.

At CATIE we employed a nested experimental design, in efforts to quantify the full extent of potential ITV of coffee leaves, thereby enhancing our understanding of ITV across different levels of biological or spatial organization [39]. We sampled coffee plants from each of the

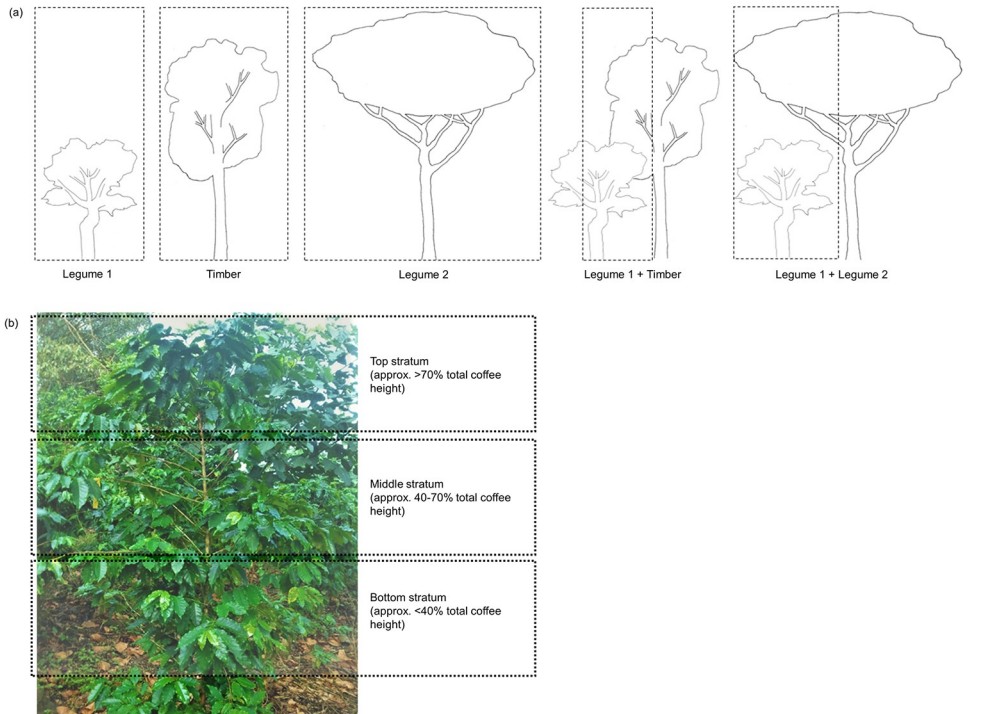

**Fig 1. Plot- and plant-level sampling requirements.** (a) The five shade tree treatments are represented: *Erythrina poeppigiana* (Legume 1), *Terminalia amazonia* (Timber), *Chloroleucon eurycyclum* (Legume 2), *E. poeppigiana + T. amazonia* (Legume 1 + Timber) and *E. poeppigiana + C. eurycyclum* (Legume 1 + Legume 2). Dotted boxes represent study plot boundaries for all measurements. (b) Coffee plant profile with each of the plant strata highlighted (top, middle, bottom) from which leaves were sampled.

three blocks at the CATIE site. Within each block, we sampled plants growing in two amendment regimes: moderate conventional (with applications of chemical soil fertilizer, foliar minerals, and foliar copper-based fungicide with cyproconazole), and intensive organic management (with applications of chicken manure soil fertilizer and foliar Bordeaux mixture fungicide).

Within each amendment regime, we sampled from five different shade tree treatments: coffee shaded under single-species canopies of *Erythrina poeppigiana* (Legume 1), *Chloroleucon eurycyclum* (Legume 2), and *Terminalia amazonia* (Timber) and shaded under double-species canopies of *E. poeppigiana + C. eurycyclum* (Legume 1 + Legume 2) and *E. poeppigiana + T. amazonia* (Legume 1 + Timber) (Fig 1A). The shade trees in this study included both timber and leguminous species, a common practice in coffee agroforestry systems in the study region of central Costa Rica [40].

Within each shade tree treatment, we sampled coffee plants that fell into one of two yield classes: a low fruiting class (0–200 fruiting nodes), and a high fruiting class (200+ fruiting nodes), as determined by counting fruiting nodes [41], reflecting a similar yield threshold observed by [36], where shade levels interact with CLR differently around this critical threshold. Within each yield class, three coffee plants were selected of similar age since stump-pruning (between 3–4 years old), as estimated by stem diameter [42]. All coffee plants were within a representative area of ≤10 m diameter, either directly beneath the canopy of a representative shade tree in single-species canopies, or beneath the overlapping canopies of representative shade trees in double-species canopies (Fig 1A). To capture potential intra-plant variability

[43], we sampled leaves from different branches within each coffee plant from top (>70% of total plant height), middle (40–70% of total plant height), and bottom (<40% of total plant height) canopy strata (Fig 1B). Within each stratum, we followed standardized leaf sampling protocols for leaf traits [8] and selected the first three recently developed, fully expanded coffee leaves from the branch tip that demonstrated signs of CLR infection for leaf trait measurements.

## Coffee leaf measurements

We measured seven morphological and chemical leaf functional traits for each sampled coffee leaf, following protocols of [8]: leaf area ($cm^2$) using ImageJ software (1.47 bundled with 64-bit Java; [44]), specific leaf area ($mg\ mm^{-2}$), leaf dry matter content (LDMC; $mg\ g^{-1}$), and leaf thickness (mm), averaged from measurements collected near the tip, middle and base of the leaf using electronic calipers. Stomatal density (number of stomates $mm^{-2}$) was quantified for one leaf per stratum by taking an impression of the middle abaxial portion of the leaf with diluted clear nail enamel [45]. A 1 $mm^2$ section of each impression was used to count stomata present through a compound microscope (100x total magnification). The three leaves from each stratum were then combined for analysis of LN ($mg\ g^{-1}$) and leaf carbon to N ratio (CN) using a LECO CHN Elemental Analyzer (St. Joseph, Michigan, USA). Our analysis was restricted to these chemical traits to reflect what is commonly reported in the functional trait literature. On each coffee leaf, CLR severity was quantified as the percent area of leaf with visible chlorotic lesions [46], determined using ImageJ software.

## Statistical analyses

All statistical analyses were performed in RStudio version 1.1.456 (R Foundation for Statistical Computing, Vienna, Austria). We first calculated means and standard errors for all leaf traits and CLR severity and used a two-way analysis of variance (ANOVA) with a Tukey's post-hoc test to compare trait and CLR means among the shade tree canopy treatments and coffee plant stratum (where $n = 519$ leaves for all variables in each plant stratum, except for stomatal density and leaf chemical traits where $n = 173$). We then examined bivariate relationships among all leaf traits and CLR severity across all shade tree treatments, but focusing only on the middle stratum (where $n = 519$ for all models, except for stomatal density and leaf chemical traits where $n = 173$) using standardized major axis (SMA) regression models implemented in the 'lmodel2' R package [47]. CLR severity, leaf thickness, SLA, LDMC and CN were not normally distributed and were therefore log-transformed as needed prior to analysis.

To understand whether or not relationships between coffee leaf functional traits and CLR severity differ between different shade tree canopy treatments and coffee plant strata, we used a hierarchical clustering analysis implemented in the 'dendextend' R package [48]. Specifically, this analysis first determined the hierarchical clustering of coffee leaf traits and CLR severity using 'Ward's' method. In a dendrogram, each tip represents an individual leaf, where tips further apart represent leaves that are more dissimilar in their traits. We limited the traits to those with significant relationships with CLR (i.e., removing stomatal density). Cluster dendrograms for each trait-by-trait combination were aligned and compared using the 'step1side' method, from which an entanglement score was then calculated. In this analysis, a higher entanglement score indicates greater dissimilarity of the clusters, and a lower score indicates a better alignment or tighter relation between the clusters.

We then merged our leaf trait data from only the middle canopy stratum, with leaf trait data from previous work [20] on non-diseased Caturra plants collected from the same site, to test for differences in ITV relationships between diseased and healthy leaves. This published data is from the same coffee plant stratum (the middle stratum), cultivar, similar management

regimes, and was collected at the same time of year, making this arguably the most comparable dataset on coffee leaf traits available. However, this data ($n = 192$) was from strictly healthy coffee plants with no signs of foliar disease [20]. To test for potential differences in bivariate leaf trait relationships between leaves with and without foliar disease, we used SMA and a linear hypothesis test using the 'smatr' package in R [49], to compare the slope and intercept of relationships between SLA-LN, and SLA-CN.

This merged dataset, including diseased and non-diseased trait values, was then further analyzed using principal components analysis (PCA) with the 'vegan' package in R [50]. Our PCA included leaf thickness, leaf area, SLA, LN and leaf CN, in order to include data that was present in both datasets. PCA results were grouped according to presence or absence of visible foliar disease, where $n = 519$ for leaves with CLR and $n = 192$ for those without. A permutational multivariate analysis of variance (PERMANOVA) implemented with the 'adonis' function in the 'vegan' R package [50] was used to test for differences in multivariate trait variation between the two groupings (with 999 permutations used).

# Results

## Intraspecific trait variation of coffee leaves under diverse canopies

Coffee leaf traits differed significantly between both the shade tree canopy treatments and coffee plant strata (Table 1). However, since there were no consistent significant differences in leaf trait values across other levels of the nested experimental design (i.e., among blocks, amendments, or plant yield classes), these were not explored in subsequent analyses. Morphological traits varied significantly between different shade tree treatments; in the middle canopy stratum, SLA was highest ($20.4 \pm 0.4$ mg mm$^{-2}$) and LDMC lowest ($235.7 \pm 2.5$ mg g$^{-1}$) in the

**Table 1. Means and standard errors of CLR severity and coffee leaf traits across shade tree treatments and coffee plant strata.**

| | Legume 1 | | | Timber | | | Legume 2 | | | Legume 1 + Timber | | | Legume 1 + Legume 2 | | |
|---|---|---|---|---|---|---|---|---|---|---|---|---|---|---|---|
| | Top | Middle | Bottom | Top | Middle | Bottom | Top | Middle | Bottom | Top | Middle | Bottom | Top | Middle | Bottom |
| **CLR** | 0.5$^{bB}$ | 0.8$^{aB}$ | 0.7$^{abB}$ | 1.2$^{bA}$ | 1.7$^{abB}$ | 2.7$^{aA}$ | 1.2$^{bA}$ | 3.6$^{aA}$ | 2.6$^{abA}$ | 0.5$^{bB}$ | 1.1$^{aB}$ | 0.8$^{abB}$ | 0.5$^{B}$ | 0.8$^{B}$ | 0.8$^{B}$ |
| | ±0.1 | ±0.1 | ±0.1 | ±0.2 | ±0.4 | ±0.4 | ±0.2 | ±0.7 | ±0.7 | ±0.1 | ±0.2 | ±0.1 | ±0.1 | ±0.1 | ±0.2 |
| **T** | 0.25$^{a}$ | 0.24$^{bB}$ | 0.22$^{cB}$ | 0.25$^{a}$ | 0.25$^{bA}$ | 0.23$^{bA}$ | 0.24$^{a}$ | 0.24$^{aB}$ | 0.22$^{bB}$ | 0.25$^{a}$ | 0.24$^{bB}$ | 0.23$^{cAB}$ | 0.25$^{a}$ | 0.24$^{abAB}$ | 0.23$^{bAB}$ |
| | ±0.003 | ±0.002 | ±0.002 | ±0.002 | ±0.002 | ±0.002 | ±0.002 | ±0.003 | ±0.002 | ±0.002 | ±0.002 | ±0.002 | ±0.003 | ±0.002 | ±0.002 |
| **S** | 139.8$^{B}$ | 133.6 | 123.9 | 164.1$^{aA}$ | 149.1$^{ab}$ | 131.9$^{b}$ | 133.7$^{B}$ | 139.5 | 129.0 | 131.2$^{B}$ | 126.9 | 123.0 | 125.3$^{abB}$ | 138.7$^{a}$ | 113.7$^{b}$ |
| | ±5.5 | ±4.4 | ±5.3 | ±6.8 | ±8.2 | ±5.0 | ±4.6 | ±5.4 | ±5.0 | ±4.7 | ±8.7 | ±3.6 | ±4.4 | ±5.1 | ±4.2 |
| **LA** | 37.5$^{BC}$ | 41.3$^{BC}$ | 39.6$^{C}$ | 34.6$^{bC}$ | 39.7$^{aC}$ | 38.7$^{aC}$ | 39.4$^{B}$ | 39.1$^{C}$ | 40.4$^{BC}$ | 47.0$^{A}$ | 47.0$^{A}$ | 45.6$^{AB}$ | 50.1$^{aA}$ | 45.3$^{bAB}$ | 49.5$^{abA}$ |
| | ±1.2 | ±1.4 | ±1.2 | ±1.3 | ±1.4 | ±1.1 | ±1.2 | ±1.1 | ±1.4 | ±1.3 | ±1.3 | ±1.5 | ±1.6 | ±1.3 | ±1.5 |
| **SLA** | 15.0$^{cB}$ | 16.3$^{bB}$ | 18.2$^{aC}$ | 15.7$^{bB}$ | 16.8$^{aB}$ | 17.2$^{aC}$ | 17.3$^{cA}$ | 18.8$^{bA}$ | 20.0$^{aB}$ | 17.1$^{bA}$ | 19.7$^{aA}$ | 19.8$^{aB}$ | 18.0$^{bA}$ | 20.4$^{aA}$ | 21.6$^{aA}$ |
| | ±0.3 | ±0.2 | ±0.3 | ±0.5 | ±0.3 | ±0.3 | ±0.3 | ±0.4 | ±0.4 | ±0.3 | ±0.3 | ±0.3 | ±0.3 | ±0.4 | ±0.5 |
| **LDMC** | 276.0$^{aA}$ | 262.0$^{bA}$ | 253.7$^{bA}$ | 276.7$^{aA}$ | 257.1$^{bAB}$ | 255.8$^{bA}$ | 250.8$^{aC}$ | 241.8$^{abC}$ | 233.4$^{bB}$ | 268.3$^{aAB}$ | 244.5$^{bBC}$ | 248.0$^{bAB}$ | 251.0$^{aBC}$ | 235.7$^{bC}$ | 237.0$^{bB}$ |
| | ±4.3 | ±3.4 | ±3.6 | ±5.0 | ±4.5 | ±4.6 | ±3.8 | ±3.9 | ±4.6 | ±3.4 | ±3.0 | ±3.3 | ±3.3 | ±2.5 | ±3.7 |
| **LN** | 3.25$^{bB}$ | 3.30$^{abB}$ | 3.38$^{aA}$ | 2.95$^{bD}$ | 3.16$^{aC}$ | 3.03$^{abC}$ | 3.07$^{bC}$ | 3.14$^{abC}$ | 3.21$^{aB}$ | 3.24$^{bB}$ | 3.31$^{abAB}$ | 3.37$^{aA}$ | 3.40$^{A}$ | 3.41$^{A}$ | 3.44$^{A}$ |
| | ±0.03 | ±0.03 | ±0.03 | ±0.04 | ±0.04 | ±0.04 | ±0.02 | ±0.02 | ±0.03 | ±0.03 | ±0.03 | ±0.03 | ±0.03 | ±0.02 | ±0.03 |
| **CN** | 14.13$^{aC}$ | 13.82$^{abB}$ | 13.43$^{bC}$ | 15.85$^{aA}$ | 14.67$^{bA}$ | 15.30$^{abA}$ | 15.01$^{aB}$ | 14.61$^{abA}$ | 14.26$^{bB}$ | 14.09$^{aC}$ | 13.74$^{abBC}$ | 13.48$^{bC}$ | 13.41$^{D}$ | 13.29$^{C}$ | 13.07$^{C}$ |
| | ±0.15 | ±0.13 | ±0.16 | ±0.25 | ±0.18 | ±0.24 | ±0.13 | ±0.12 | ±0.15 | ±0.12 | ±0.13 | ±0.15 | ±0.13 | ±0.11 | ±0.14 |

CLR = coffee leaf rust severity (%); T = leaf thickness (mm); S = stomatal density (number of stomates mm$^{-2}$); LA = leaf area (cm$^2$); SLA = specific leaf area (mg mm$^{-2}$); LDMC = leaf dry matter content (mg g$^{-1}$); LN = leaf nitrogen concentration (mg g$^{-1}$); CN = leaf carbon to nitrogen ratio. Lower case letters beside mean values denote significant differences ($P < 0.05$) between strata within a single shade tree treatment, whereas upper case letters denote significant differences ($P < 0.05$) between shade tree treatments within a single stratum, as determined by a two-way ANOVA with post-hoc Tukey HSD test.

Legume 1 + Legume 2 treatment. In contrast, SLA was lowest (16.3 ± 0.2 mg mm$^{-2}$) and LDMC the highest (262.0 ± 3.4 mg g$^{-1}$) in the Legume 1 treatment. Chemical traits varied according to the presence or absence of Legume 1 (*E. poeppigiana*), such that LN was significantly higher and CN significantly lower when Legume 1 was present across all coffee plant strata (ANOVA $P < 0.001$, Table 1).

Across all shade tree treatments, SLA (16.66 ± 0.15 mg mm$^{-2}$) and LN (3.19 ± 0.01 mg g$^{-1}$) were lowest and LDMC (264.12 ± 1.83 mg g$^{-1}$) and CN (14.44 ± 0.08) were highest in the top coffee stratum ($P < 0.001$). Only under Legume 1 + Legume 2 were there no significant differences observed in LN and CN across strata. CLR severity was lower in the top stratum and higher in the middle stratum, except for under Timber, where CLR severity was highest in the bottom stratum (Table 1).

## Leaf trait co-variation and disease severity

In the middle canopy stratum, and across all shade tree treatments, we found evidence of coordinated trait relationships and trait co-variation with CLR severity (Table 2). Specifically, we found both greater SLA and LN were related to lower CLR severity ($r^2 = 0.03$, $P < 0.001$, and $r^2 = 0.078$, $P < 0.001$, respectively), whereas greater LDMC and CN were related to greater CLR severity ($r^2 = 0.019$, $P = 0.002$, and $r^2 = 0.059$, $P < 0.001$, respectively). Relationships between traits and CLR were robust when analyzing each canopy plant stratum individually, except for the top stratum where SLA and LDMC were not significantly related to CLR (S1 Table).

## Strength of trait-disease relationships

Our hierarchical clustering analyses and entanglement scores indicated that differences in the strength of relationships between CLR severity and six leaf traits (specifically leaf thickness, leaf area, SLA, LDMC, LN, CN). Across each stratum, and across all shade tree treatments,

**Table 2. Bivariate regressions of CLR severity and coffee leaf traits across all shade tree treatments in the middle stratum.**

| | Disease metric | Morphological leaf traits | | | | | Chemical leaf traits | |
|---|---|---|---|---|---|---|---|---|
| | log CLR | log T | S | LA | log SLA | log LDMC | LN | log CN |
| log CLR | – | 6.56 | -0.016 | -0.050 | -3.57 | 4.52 | -2.24 | 6.55 |
| | | (6.02, 7.15) | (-0.019, -0.014) | (-0.054, -0.046) | (-3.89, -3.28) | (4.15, 4.92) | (-2.43, -2.06) | (6.02, 7.12) |
| log T | **0.02** | – | -2.60E-3 | 0.008 | -0.54 | 0.69 | 0.34 | -1.00 |
| | **(<0.001)** | | (-0.003, -0.002) | (0.007, 0.008) | (-0.59, -0.50) | (0.63, 0.75) | (0.31, 0.37) | (-1.09, -0.92) |
| S | 0.006 | 0.004 | – | -2.90 | -215.96 | 296.36 | -129.75 | 379.29 |
| | (0.304) | (0.404) | | (-3.35, -2.50) | (-250.92, -185.86) | (256.16, 342.87) | (-150.57, -111.80) | (327.04, 439.90) |
| LA | **0.017** | **0.054** | **0.046** | – | 71.98 | -91.11 | 45.15 | -131.95 |
| | **(0.003)** | **(<0.001)** | **(0.005)** | | (66.14, 78.34) | (-99.26, -83.63) | (41.56, 49.06) | (-143.32, -121.48) |
| log SLA | **0.030** | **0.101** | 0.004 | **0.038** | – | -1.27 | 0.63 | -1.83 |
| | **(<0.001)** | **(<0.001)** | (0.390) | **(<0.001)** | | (-1.35, -1.180 | (0.58, 0.68) | (-1.99, -1.69) |
| log LDMC | **0.019** | 0.006 | **0.061** | **0.014** | **0.406** | – | -0.50 | 1.45 |
| | **(0.002)** | (0.080) | **(0.001)** | **(0.007)** | **(<0.001)** | | (-0.54, -0.46) | (1.34, 1.57) |
| LN | **0.078** | 0.004 | 0.020 | **0.076** | **0.088** | **0.186** | – | -2.92 |
| | **(<0.001)** | (0.142) | (0.062) | **(<0.001)** | **(<0.001)** | **(<0.001)** | | (-2.97, -2.88) |
| log CN | **0.059** | 0.006 | **0.029** | 0.082 | 0.074 | 0.185 | 0.970 | – |
| | **(<0.001)** | (0.078) | **(0.026)** | **(<0.001)** | **(<0.001)** | **(<0.001)** | **(<0.001)** | |

See Table 1 for trait abbreviations. Model $r^2$ and $P$-values (in brackets) for each bivariate standardized major axis regression analysis are presented in the lower left section of the matrix, with significant relationships ($P < 0.05$) highlighted in bold. The upper right section of the matrix presents model slopes and associated 95% confidence intervals (in brackets).

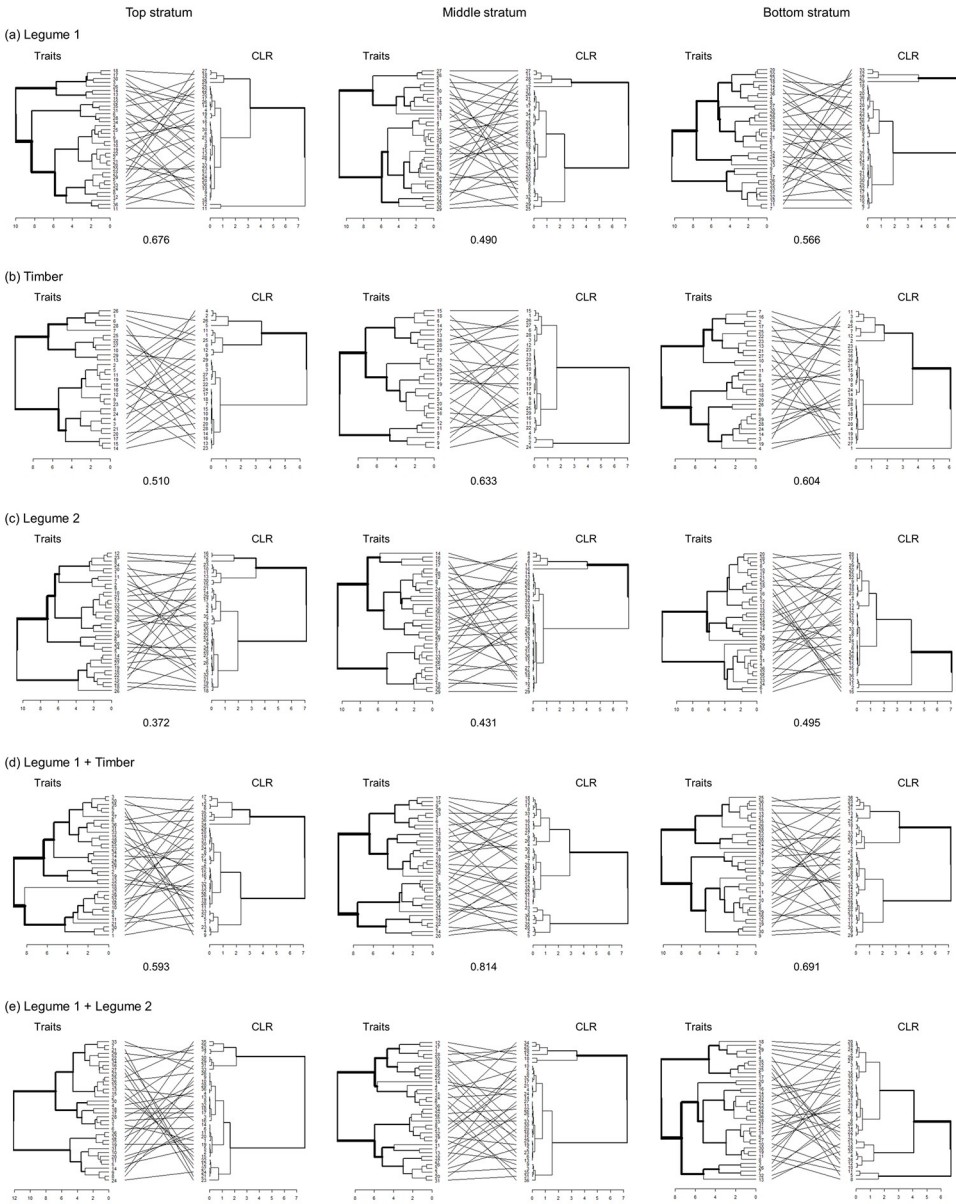

**Fig 2. Paired dendrograms of coffee traits and CLR severity by plant strata.** (a) *Erythrina poeppigiana* (Legume 1), (b) *Terminalia amazonia* (Timber), (c) *Chloroleucon eurycyclum* (Legume 2), (d) *E. poeppigiana* + *T. amazonia* (Legume 1 + Timber), and (e) *E. poeppigiana* + *C. eurycyclum* (Legume 1 + Legume 2). Entanglement scores are listed below each paired dendrogram. Euclidean distances were clustered using Ward's method. Paired dendrograms were first untangled to find the best alignment layout.

entanglement scores ranged nearly 2-fold from 0.402 to 0.709 (Fig 2). Across all shade treatments pooled together, there was stronger alignment between traits and CLR severity in the middle stratum, except for Legume 1, where the top stratum had the strongest alignment (Fig 2A).

## Differences in diseased and healthy plants

Patterns of coffee trait co-variation differed significantly between diseased and healthy plants (Fig 3). Using leaf trait data from only the middle coffee plant stratum, trait relationships

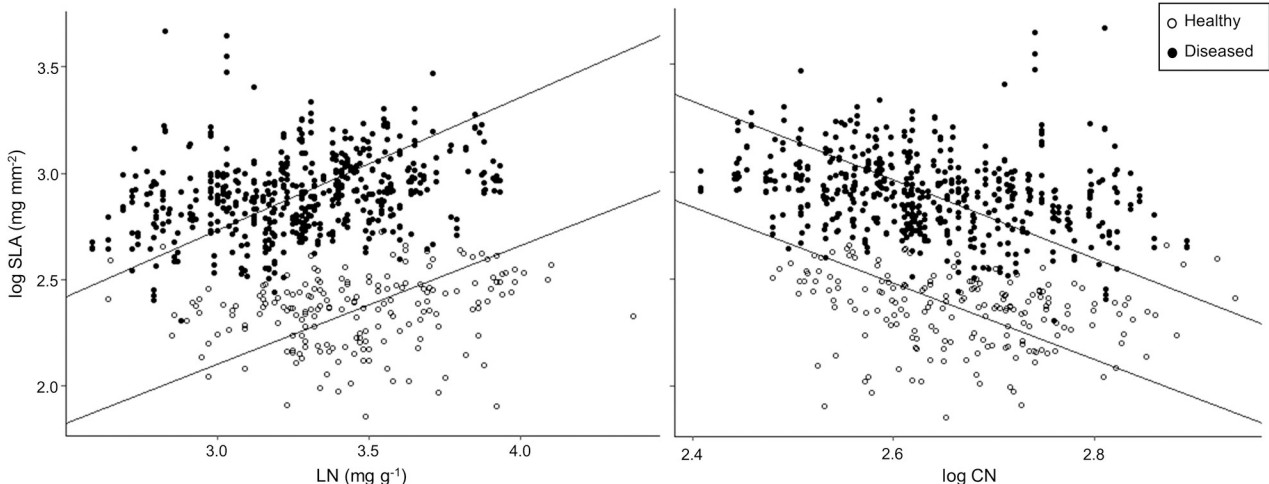

**Fig 3. Key bivariate trait relationships from healthy and diseased leaves from the middle coffee plant stratum.** (A) Log-transformed specific leaf area (SLA) and leaf nitrogen concentration (LN), and (B) log-transformed SLA and log-transformed leaf carbon to nitrogen ratio (CN), where $n = 519$ for diseased leaves and $n = 192$ for healthy leaves.

shifted in relation to the presence or absence of CLR. Specifically, we found that bivariate relationships between SLA and LN was stronger in diseased leaves (SMA $r^2 = 0.088$; $P < 0.001$) with a significantly steeper slope ($0.63 \pm 0.02$; $P = 0.006$) and larger intercept ($0.85 \pm 0.03$; $P < 0.001$) compared with healthy leaves (SMA $r^2 = 0.024$; $P < 0.001$; slope = $0.56 \pm 0.03$; intercept = $0.43 \pm 0.05$) (Fig 3). Similarly, we observed a stronger relationship between SLA and CN in diseased leaves (SMA $r^2 = 0.074$; $P < 0.001$) with a significantly greater intercept ($7.73 \pm 0.20$; $P = 0.003$) compared to healthy leaves (SMA $r^2 = 0.041$; $P = 0.005$; intercept = $7.11 \pm 0.33$).

Principal component analysis (PCA) of our merged dataset of leaves with and without visible signs of foliar disease, indicated that the first two PCA axes explain a total of 70.4% of the variation in the coffee leaf functional traits included (Fig 4). PCA axis 1 explained 40.7% of variation in multivariate traits being significantly positively associated with LN ($r^2 = 0.835$, $P < 0.001$), leaf area ($r^2 = 0.204$, $P < 0.001$) and SLA ($r^2 = 0.068$, $P < 0.001$), and negatively associated with CN ($r^2 = 0.923$, $P < 0.001$). PCA axis 2 explained an additional 29.7% of the variation in traits and was significantly positively associated with leaf thickness ($r^2 = 0.741$, $P < 0.001$) and LN ($r^2 = 0.045$, $P < 0.001$), and negatively associated with SLA ($r^2 = 0.687$, $P < 0.001$). PERMANOVA indicated that leaf traits in our merged dataset differed significantly between healthy and diseased environments ($F = 124.86$, $P < 0.001$), such that diseased leaves expressed greater SLA, whereas healthy leaves shifted towards greater leaf thickness.

## Discussion

### Intra-plant trait variability

While multiple studies have documented intraspecific trait variation in coffee and other wild and domesticated plant species, e.g. [2,24,51], our results demonstrate for the first time that ITV and trait relationships are maintained in leaves with visible signs of foliar disease. Furthermore, we found evidence of intra-plant trait variability across three coffee plant strata independent of shade tree treatment, which demonstrates that caution should be taken when reporting trait mean values to characterize an entire plant. Specifically, we found that leaves from the bottom stratum expressed more resource acquiring trait values (i.e., high SLA coupled with

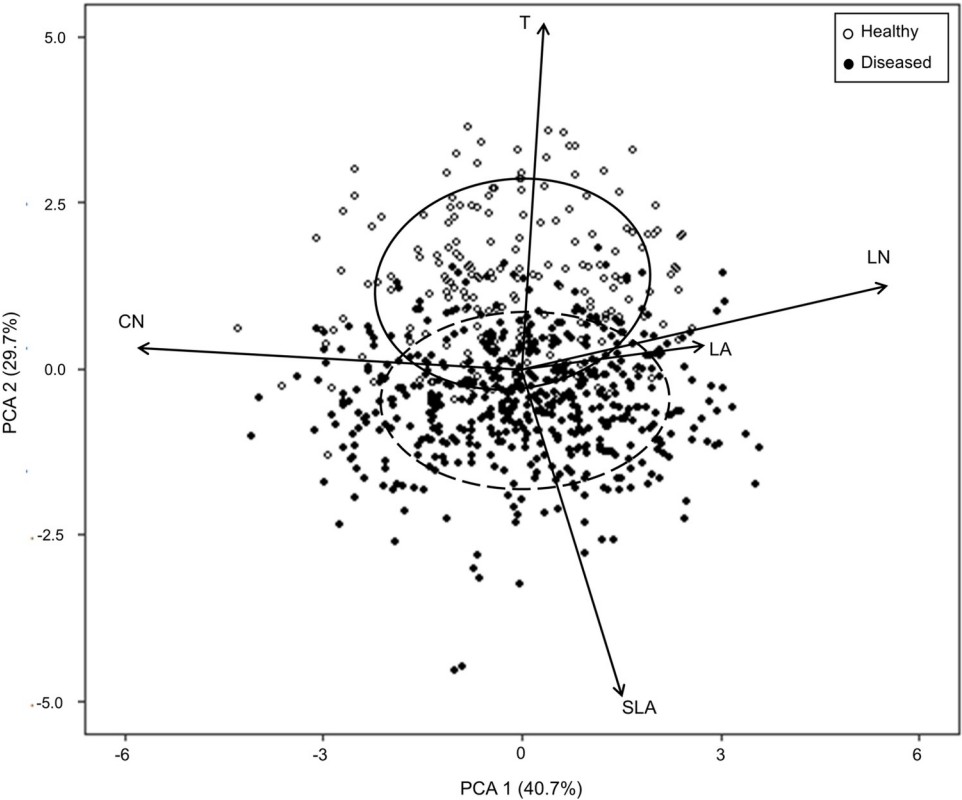

**Fig 4. Principal component analysis of five coffee leaf functional traits from the middle stratum.** Traits include leaf thickness (T), leaf area (LA), specific leaf area (SLA), leaf nitrogen concentration (LN), and leaf carbon to nitrogen ratio (CN). Symbols represent presence or absence of CLR (where $n = 519$ for diseased leaves and $n = 192$ for healthy leaves). Circles correspond to 95% confidence ellipses for coffee leaves sampled within each category, where the solid line represents the healthy grouping, and the dashed line represents the diseased grouping.

high LN), whereas leaves from the top stratum expressed resource conserving trait values (i.e., high LDMC coupled with high leaf CN). This is similar to findings by [52], who found that leaf area, SLA and LN were highest in the bottom coffee plant stratum.

This stratification was also evident in the alignment of key coffee leaf functional traits and CLR severity through hierarchical clustering. We found that coffee leaf trait clusters (specifically based on leaf thickness, leaf area, SLA, LDMC, LN, CN) were more strongly aligned with CLR severity in the middle stratum; a finding that was consistent across all but one shade tree treatment. These patterns can be the result of the interacting role of light availability on coffee leaf trait expression. In coffee, leaf trait plasticity in part is driven by the heterogeneity of light transmittance [17], which leads to greater resource acquiring trait values in light-limited environments [24], and/or self-shaded leaves [53]. These patterns can also be a consequence of differences in intra-plant uredospore deposition patterns, where more spores are deposited on lower coffee plant strata [54], leading to greater CLR severity in lower strata, as observed in our study. Quantifying these complex trait-environment-pathogen interactions are therefore vital in understanding and modelling pathological dynamics in agroecosystems.

## Leaf functional traits and foliar disease

While CLR incidence depends on the amount of inoculum present in a given area [55], the severity to which individual coffee leaves are affected by CLR depends on foliar traits that

could change in response to, or interaction with, the pathogen. We hypothesized a positive relationships between stomatal density and leaf susceptibility to CLR, due to the increased probability of uredospores penetrating through stomatal openings [27]. However, in our study we did not find evidence of a significant relationship between CLR severity and stomatal density, with little variability of stomatal density across the different shade tree treatments and coffee plant strata. Therefore, CLR severity is likely more closely related to actual stomatal functioning (i.e., stomatal opening), as it could be the opening of stomata that promotes the increased release of sporogenous pathogen cells [56]. In this case, variability in stomatal opening and pathogen behaviour is hypothesized to be related to key microclimate variability with specific timing in the pathogen's lifecycle [33].

In the middle coffee canopy stratum, we found that more resource conservative trait values (i.e., high LDMC and high leaf CN values) were positively correlated with CLR severity, whereas greater resource acquisitive trait values (i.e., high SLA and high LN values) were significantly negatively correlated across all levels of the nested experimental design, contradicting our second hypothesis. Based on previous research, one would expect that leaves with more resource acquisitive trait values would have greater CLR severity, as larger leaf area is thought to increase the probability of *H. vastatrix* uredospore interception [27], and studies have demonstrated a positive correlation between SLA and disease severity in other plant species [12], potentially due to thinner cell walls and more non-structural carbohydrates aiding biotrophic fungal pathogens [12]. When considering nutrient-based traits, there are more diverging results in the literature. For example, high N supply (via fertilization of the soil) combined with low light (similar to the shaded conditions of our study), presumably leading to greater foliar N content, resulted in slower rates of infection and sporulation of *H. vastatrix* in *C. arabica* [57]. In contrast, limited soil N availability for wheat resulted in reduced spore production of leaf rust (*Puccinia triticina*) due to less efficient sporulation [58]. In Douglas-fir infected with the biotrophic fungus *Phaeocryptopus gaeumannii*, high LN is correlated with high disease severity, likely due to greater nutrient availability for the pathogen within the apoplast, where *P. gaeumannii* resides, and from potential cell membrane leakage [14]. These contrasting results are expected, given the unique nature of different plant-pathogen co-evolution strategies and, in turn, the pathogen-specific effect of foliar nutrient availability on disease, dictating both the level of defensive compound production and the metabolic activity of host cells [13,59].

Our results on the individual relationships between traits and disease severity, though significant, were weak, highlighting how analysing traits individually can provide only some insight into potential constitutive or induced leaf-level mechanisms. Rather, considering traits together as trait syndromes better reflects plant resource use strategies specific to the plant genotype in question. Through this lens, our results suggest that the greater resource acquisitive traits, which together promote C assimilation rates in coffee [24], could lead to reduced plant stress and improved disease tolerance via compensatory growth [15,60]. Indeed, changes in C assimilation rates have reflected a key tolerance mechanism in other pathogen-tolerant plant genotypes, such as *Senecio vulgari* (common groundsel) with the fungal pathogen *Coleosporium tussilaginis* [61]. Given that CLR severity was measured as the percent leaf area with visible chlorotic lesions [46], enhanced leaf growth can lead to a dilution effect [62], reducing the magnitude of CLR severity measured, and should be considered in future studies.

### Intraspecific leaf functional trait relationships in diseased plants

When comparing trait relationships in leaves with and without visible signs of foliar disease, we found that the shape of these relationships remained consistent, highlighting that CLR does

not significantly change established global leaf trait relationships. However, despite general consistencies, there was an overall shift towards higher resource acquiring trait values when foliar disease was apparent, confirming our third hypothesis. Based on PCA results, coffee leaves that showed signs of CLR shifted significantly towards greater SLA values along PCA axis 2, compared to healthy coffee leaves with no signs of foliar disease that shifted towards greater leaf thickness. Similarly, key bivariate relationships between SLA with LN and CN shifted towards greater SLA values for a given LN or CN value, in diseased compared to healthy leaves. As noted above, previous research has suggested that greater resource acquiring trait values, including high SLA, are linked with greater disease susceptibility due to structural foliar differences [12]. Therefore, this shift towards resource acquisition in diseased leaves likely reflects a greater constitutive susceptibility to pathogen infection and development, rather than an induced response to pathogen infection [63].

## Conclusion

Our results provide evidence that plant functional traits and their relationships shift under pathogen pressures, representing among the first lines of evidence indicating that ITV plays a role in mediating disease severity, specifically coffee leaf rust in a managed agroecosystem. In doing so, our work both advances our understanding of the functional ecology of a major crop pathogen and addresses a vital gap in our understanding of global patterns of functional trait variation. Overall, we offer first insights into the potential role of ITV in plant-pathogen-environment interactions, still only a glimpse into the response-effect co-evolution between plants and their native pathogens, highlighting the need to further explore foliar pathogens within the global functional trait space.

## Supporting information

**S1 Checklist. Inclusivity in global research checklist.**
(PDF)

**S1 Table. Bivariate regressions of CLR severity and coffee leaf traits across all shade tree treatments in the (a) top stratum and (b) bottom stratum.**
(PDF)

## Acknowledgments

The authors sincerely thank L. Romero and V. H. Mendez Sanabria for their invaluable assistance in the field, and the AGROFORESTA Scientific Platform for its support and collaboration with CATIE's Long-Term Trial with Agroforestry Systems in Cafe. We would also like to thank the anonymous reviewers for reading earlier versions of this paper and providing helpful and constructive comments.

## Author Contributions

**Conceptualization:** Stephanie Gagliardi, Jacques Avelino, Marney E. Isaac.

**Formal analysis:** Stephanie Gagliardi.

**Investigation:** Stephanie Gagliardi.

**Methodology:** Stephanie Gagliardi, Jacques Avelino, Marney E. Isaac.

**Resources:** Jacques Avelino, Elias de Melo Virginio Filho, Marney E. Isaac.

**Supervision:** Jacques Avelino, Marney E. Isaac.

**Writing – original draft:** Stephanie Gagliardi, Jacques Avelino, Adam R. Martin, Marc Cadotte, Elias de Melo Virginio Filho, Marney E. Isaac.

**Writing – review & editing:** Stephanie Gagliardi, Jacques Avelino, Adam R. Martin, Marc Cadotte, Elias de Melo Virginio Filho, Marney E. Isaac.

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
