## [Decision Letter · Decision Letter 0]

30 Jan 2023

PONE-D-22-31465Leaf functional traits and pathogens: how foliar disease shapes intraspecific trait variation in agroecosystemsPLOS ONE

Dear Dr. Isaac,

Thank you for submitting your manuscript to PLOS ONE. After careful consideration, we feel that it has merit but does not fully meet PLOS ONE’s publication criteria as it currently stands. Therefore, we invite you to submit a revised version of the manuscript that addresses the points raised during the review process.

We look forward to receiving your revised manuscript.

Kind regards,

Lukas Beule

Academic Editor

PLOS ONE

Journal Requirements:

2. In your Methods section, please provide additional information regarding the permits you obtained for the work. Please ensure you have included the full name of the authority that approved the field site access and, if no permits were required, a brief statement explaining why

3. Please include a complete copy of PLOS’ questionnaire on inclusivity in global research in your revised manuscript. Our policy for research in this area aims to improve transparency in the reporting of research performed outside of researchers’ own country or community. The policy applies to researchers who have travelled to a different country to conduct research, research with Indigenous populations or their lands, and research on cultural artefacts. The questionnaire can also be requested at the journal’s discretion for any other submissions, even if these conditions are not met.  Please find more information on the policy and a link to download a blank copy of the questionnaire here: https://journals.plos.org/plosone/s/best-practices-in-research-reporting. Please upload a completed version of your questionnaire as Supporting Information when you resubmit your manuscript

"This work was supported by the Natural Sciences and Engineering Research Council of Canada (Discovery Grant to M.E.I. and Alexander Graham Bell Canada Graduate Scholarship to S.G.)."

"This work was supported by the Natural Sciences and Engineering Research Council of Canada (Discovery Grant to M.E.I. and Alexander Graham Bell Canada Graduate Scholarship to S.G.). The funders had no role in study design, data collection and analysis, decision to publish, or preparation of the manuscript."

Additional Editor Comments:

Dear Authors,

I received two Review Reports for the manuscript entitled "Leaf functional traits and pathogens: how foliar disease shapes intraspecific trait variation in agroecosystems", which you submitted to PLOS ONE. Based one the Review Reports received, I have decided that your manuscript could be reconsidered for publication should you be prepared to incorporate minor revisions. Please carefully check the comments of the Reviewers, especially those of Reviewer 2.

Thank you for submitting to PLOS ONE. I look forward to receiving your revised manuscript.

With kind regards,

Lukas Beule (Academic Editor)

Reviewers' comments:

Reviewer's Responses to Questions

**Comments to the Author**

1. Is the manuscript technically sound, and do the data support the conclusions?

Reviewer #1: Partly

Reviewer #2: Yes

2. Has the statistical analysis been performed appropriately and rigorously? 

Reviewer #1: Yes

Reviewer #2: Yes

3. Have the authors made all data underlying the findings in their manuscript fully available?

Reviewer #1: Yes

Reviewer #2: Yes

4. Is the manuscript presented in an intelligible fashion and written in standard English?

Reviewer #1: Yes

Reviewer #2: Yes

5. Review Comments to the Author

Reviewer #1: Audacious title that does not reflect the experiment conducted..... I suggest modifying it for the evaluated pathosystem that is Coffea arabica x Hemileia vastatrix.

Figure 2 should be redone.

The studied species Erythrina poeppigiana, Chloroleucon eurycyclum, and Terminalia amazonia are commonly used in intercropping with coffee trees by coffee growers????

With regard to the nutritional part, which has 14 nutrients, why did you discuss only nitrogen?

Due to the large number of parameters, the manuscript as presented is confusing .... I suggest reformulation....

Reviewer #2: Dear Stephanie and colleagues,

I suggest the manuscript to be published after minor revisions have been completed. Specific recommendations I have to the manuscript can be found in the attached pdf of your manuscript.

The manuscript is well written and concise. It uses appropriate statistics and fits well into research working with intraspecific trait variation (ITV) and trait spectra, specifically the leaf economics spectrum (LES). Testing the impact of an obligate biotrophic leaf parasite on ITV and LES is important to get a more realistic view how trait spectra might be shaped or shifted in natrual settings where pathogens are always present, and on the other hand how they might influence disease severity. The manuscript is thus opening up new research routes within this research field.

As a background: I am not a specialist in this specific ecological research field, but a trained botanist/mycologist interested in diversity and eco-evolutionary research questions and a phytopathologist with a strong focus on rust fungi. Reading this paper and the essential "foundation papers" in the field as given in the introduction of the manuscript I think it is a very intersting approach looking at traits, intraspecific variation etc globally on a broad spectrum of plant species and responses of the plants to environmental settings, and in your case including pathogens. And once again, my applause for that.

From a phytopathological view I have many open questions, and think the paper is too plant-centristic and not giving enough credit on the active nature of plant parasites in manipulating plant species, coping with plant defense or plant traits. E.g. rust species are often very species specific, but there are ca. 8000 descirbed species of rust fungi infecting ferns, gymnosperms and angiosperms. Thus, the group as a whole copes with all kinds of leaf architecture (from minute acacia leaflets over broad-leaved soft to succulent leaves, fern leaves or fir needles (and even wood), secondary compouds etc.

Some of the papers you cite (and a plethora of additional out there) are providing an overview of the various plant-parasite-enviroment relationship (e.g. Avelino et al 2004 (citation 28). I missed broader interpreations of your research looking at the "other side" as disease severity is an outcome of what happens between the two partner given a specific environment and environmental conditions at a given time. However, I admit that this is very complicated, and therefore your approach sticking closely to the reserach field and hypotheses of the research field is probably the best approach to broaden and teset the "ITV-LES" hypothesis including plant pathogens.

So despite I doubt several of the interpreations (as what might be the causal background of the observed outcome) I have only commented on a few and deleted suggestions to make major revisions. Because otherwise the paper might have lost its clear focus for your main audience. Nevertheless, I cannot keep myself from suggesting to include the "pathogen view" in coming papers of this research topic more strongly.

6. PLOS authors have the option to publish the peer review history of their article (what does this mean?). If published, this will include your full peer review and any attached files.

Reviewer #1: No

Reviewer #2: No

---

## [Author Response · Author response to Decision Letter 0]

15 Mar 2023

Reviewer 1

1. Audacious title that does not reflect the experiment conducted..... I suggest modifying it for the evaluated pathosystem that is Coffea arabica x Hemileia vastatrix.

- Thank you for bringing up this point to make our title more specific. We have now added in the foliar disease we assess (coffee leaf rust) and changed the wording in our title (e.g., rather than “shapes”, we use “linking”). The title has now changed to “Leaf functional traits and pathogens: Linking coffee leaf rust with intraspecific trait variation in diversified agroecosystems”

2. Figure 2 should be redone.

- Given the poor image quality of Figure 2 in our original submission, we’ve re-uploaded the image file to be clearer.

3. The studied species Erythrina poeppigiana, Chloroleucon eurycyclum, and Terminalia amazonia are commonly used in intercropping with coffee trees by coffee growers????

- Indeed, intercropped tree species in coffee systems vary greatly around the world. To provide the reviewer with some clarification, we do not state that these specific species are commonly used (though E. poeppigiana is particularly common in the study region); rather, we state that the integration of timber and leguminous species in coffee systems is a common practice. To provide further clarification, we now state “The shade trees in this study included both timber and leguminous species, a common practice in coffee agroforestry systems in the study region of central Costa Rica” (lines 148-150).

4. With regard to the nutritional part, which has 14 nutrients, why did you discuss only nitrogen?

- This is a great question. Since the objective of our study is related to traits within the Leaf Economics Spectrum, we limited the analyses of chemical traits to reflect what is commonly reported in the functional trait literature. We now clarify this in lines 183-184 where we say, “Our analysis was restricted to these chemical traits to reflect what is commonly reported in the functional trait literature.”

5. Due to the large number of parameters, the manuscript as presented is confusing .... I suggest reformulation.... 

- Unfortunately, without details on which aspects of the manuscript are confusing or in which way, reformulation of the manuscript would be difficult and was not completed for these revisions.

Reviewer 2

From a phytopathological view I have many open questions, and think the paper is too plant-centric and not giving enough credit on the active nature of plant parasites in manipulating plant species, coping with plant defense or plant traits… Nevertheless, I cannot keep myself from suggesting including the "pathogen view" in coming papers of this research topic more strongly.

- Overall, we agree that this is a plant-centric paper, given the focus is on leaf ITV. We had tried to allude to the active role of pathogens, and in our revised draft, we have made this clearer throughout the manuscript. For example:

 -- Introduction, line 62, we now state “…despite emerging research linking the two and the co-evolution of many plant-pathogen systems.”

 -- Model species, lines 103-104, we now state “… a foliar disease caused by Hemileia vastatrix, a biotrophic fungus that has co-evolved with Coffea spp., …”

 -- Model species, lines 123-124, we now state “…despite the role trait expression may play in moderating plant-pathogen dynamics or the role that disease may play in manipulating common trait expression.”

 -- Discussion, lines 373-376, we now state “These contrasting results are expected, given the unique nature of different plant-pathogen co-evolution strategies and, in turn, the pathogen-specific effects of foliar nutrient availability on disease, dictating both the level of defensive compound production and the metabolic activity of host cells.”

 -- Conclusion, lines 411-412, we now state “…still only a glimpse into the response-effect co-evolution between plants and their native pathogens…”

Minor changes

- We have accepted all minor wording changes throughout the manuscript as suggested in the pdf (and noted in the “Revised Manuscript with Track Changes” file), except for the following:

 - Abstract, line 33, it was recommended to change the wording from “key coffee leaf functional traits” to “key functional traits of coffee leaves”. We have decided to maintain the original wording, as this is the accepted wording in the functional trait literature (now line 29)

 -- Introduction, line 61, it was recommended to remove “ecosystems”, as an ecosystem cannot be considered “healthy”, which we agree. However, rather than removing “ecosystem” entirely, we’ve changed the sentence to now read “… “healthy” plants or unstressed ecosystems…” to better reflect our original thought (now lines 52-53)

 -- Discussion, lines 401 (and elsewhere), it was recommended to simply state “healthy” or diseased” leaves. We have decided to keep phrasing like “with no signs of foliar disease”, as we think it is important to highlight that we are specifically discussing visible signs of disease and assuming that the coffee plants in our study could have had other diseases that we did not (and would likely have been impossible to) control for. 

Other changes

Introduction, line 72: In contrast to rust fungi, which are obligate biotrophs (and nearly exclusively) leaf parasites Ophiostoma ulmi, the causal agent of the Dutch elm disease is not an obligate biotroph. It is also not a leaf parasite, but spread by bark beetles into the wood of elm trees. This will need to be corrected. Delete citation. Potentially add another more fitting citation.

- Thank you for pointing out this discrepancy. We have removed the citation in question (Ďurkovič et al., 2013), and have adjusted the sentence as needed to reflect the reference included (Toome et al., 2010), which focused on the leaf rust Melampsora epitea. This change is now in lines 63-64.

Introduction, lines 76-79: literature backing this should be added here.

- Indeed, the structure of the paragraph made this unclear. The references we used to support the statement were in the previous paragraph. We have now moved this sentence to the previous paragraph and changed the wording to better clarify. It now reads “Although these preliminary findings are specific to plant genotypes and pathogen types, they indicate that…” (now lines 67-68).

Materials and methods, lines 104-133 (re: “Model species” subsection): I suggest to move this part to introduction still

- We moved this subsection to now be in the Introduction and have made some edits to ensure no repetition.

Model species subsection, line 115: But despite this extensive work on the biology of H. vastatrix only relatively recently surprising finding of cryptosexuality, which has a significant impact on the ability of the species to overcome resistances in the plant host and should be cited here. Carvalho, C. R., Fernandes, R. C., Carvalho, G. M., Barreto, R. W., & Evans, H. C. (2011). Cryptosexuality and the genetic diversity paradox in coffee rust, Hemileia vastatrix. PLoS One, 6(11), e26387

- We now include the suggested reference (now in line 105)

Model species subsection, line 117: This is misleading. Teliospores are also ephemerally produced (which then produce basidiospores). However, despite ample attempts to infect coffee plants with basidiospores this could never been achieved suggesting that Hemileia is a heteroecious rust (with unknown alternate host species still to be found in its centre of origin). Thus, urediniospores are the only important spore stage for the infection of coffee plants and dispersal of the coffee rust disease. But basidispospores are also "infectious units" within the life cycle of this rust species. We just do not know which plant species they infect.

- Thank you for highlighting this misleading phrasing. We now state “…the only important infectious unit of H. vastatrix of coffee…” (now line 108)

Model species subsection, line 123: To my knowledge no effect of water status of leaves (or leaf cells) has been shown for rust infection.

- You are correct. We were originally referring to the water status of leaves in relation to stomatal action. However, the role of stomatal opening in the infection cycle of H. vastatrix is still hypothetical, and so we have decided to take out this latter part of the sentence, now stating “…suggests that greater stomatal density increases H. vastatrix incidence” (now lines 113-114) 

Materials and methods, lines 163-165: This seems like a very coarse frame, but might serve the purpose, because it allows comparison with citation 35!? However, in general one would expect that finer grades of fruiting classes would be more appropriate.

- Indeed, this is a coarse frame, yet rather than selecting this frame to be comparable to the reference cited, we chose this coarse frame to reflect the critical threshold observed in the reference cited. To better explain this, we now provide clarification in lines 159-162, “Within each shade tree treatment, we sampled coffee plants that fell into one of two yield classes: a low fruiting class (0-200 fruiting nodes), and a high fruiting class (200+ fruiting nodes), as determined by counting fruiting nodes [41], reflecting a similar yield threshold observed by [36], where shade levels interact with CLR differently around this critical threshold.”

Materials and methods, lines 173-175: Please give the rationale for this.

- Our leaf sampling technique is based on standardized sampling protocols. We now include this clarification and the appropriate citation in lines 170-172, stating, “Within each stratum, we followed standardized leaf sampling protocols for leaf traits [8], and selected the first three recently developed, fully expanded coffee leaves from the branch tip that demonstrated signs of CLR infection for leaf trait measurements.”

Materials and methods, line 185: 10x total or 100x total magnification?

- It was 100x total magnification, and we’ve made the change, now in line 181.

Materials and methods, line 187: Please give manufacturer, city, country. Also for ImageJ at the end of the sentence.

- We now include the manufacturer information for the elemental analyzer (now line 183). For ImageJ, since no such information is available, we include the suggested reference for this free software (now reference #44).

Discussion, line 390: The biology of Ophiostoma ulmi and the powdery mildew of barley mentioned here are very different to rusts. While Ophiostoma is not an obligate biotroph, but has a close relation with the vectoring insects and growing in wood, powdery mildews are also obligate biotrophs but only infecting epidermal cells via direct penetration of the cuticula and epidermal cell wall and developing haustoria. A reference to literature on rust fungi would help.

- Indeed, given the focus of this paper on a leaf rust, references and comparisons should also be focused on leaf rust species. We’ve now changed the references here and the information included to reflect the new reference. We now state (now lines 383-386), “Indeed, changes in C assimilation rates have reflected a key tolerance mechanism in other pathogen-tolerant plant genotypes, such as Senecio vulgari (common groundsel) with Coleosporium tussilaginis [61]”, where reference #61 is https://doi.org/10.1094/PHYTO-96-0718

Conclusion, line 417: Please see my alternative text. I hope you can follow to phrase it a bit "softer". This less strong phrasing of the impact of ITV on the outcome of infection is a result of my view and knowledge of plant pathogens and rust fungi specifically. The way you end the paper (and indeed much of the paper) has a very plant centric approach. However, rusts in general and here, Hemileia vastatrix specifically, also very actively interact with the plants and have a plethora of evolutionary-ecological adaptations to do this successfully. Therefore, I suggest to keep the conclusions a bit less rigorous with respect of the plant traits governing or driving the outcome of the severity of the disease. 

- We have modified the conclusion following your suggestions. It now reads (in its entirety, now lines 406-413), “Our results provide evidence that plant functional traits and their relationships shift under pathogen pressures, representing among the first lines of evidence indicating that ITV plays a role in mediating disease severity, specifically coffee leaf rust in a managed agroecosystem. In doing so, our work both advances our understanding of the functional ecology of a major crop pathogen and addresses a vital gap in our understanding of global patterns of functional trait variation. Overall, we offer first insights into the potential role of ITV in plant-pathogen-environment interactions, still only a glimpse into the response-effect co-evolution between plants and their native pathogens, highlighting the need to further explore foliar pathogens within the global functional trait space.”

---

## [Editor Report · Decision Letter 1]

27 Mar 2023

Leaf functional traits and pathogens: Linking coffee leaf rust with intraspecific trait variation in diversified agroecosystems

PONE-D-22-31465R1

Dear Dr. Isaac,

We’re pleased to inform you that your manuscript has been judged scientifically suitable for publication and will be formally accepted for publication once it meets all outstanding technical requirements.

Kind regards,

Lukas Beule

Academic Editor

PLOS ONE

Additional Editor Comments (optional):

Dear authors,

you did a great job revising your manuscript. I am happy to accept it for publication.

Kind regards,

Lukas Beule
---

## [Editor Report · Acceptance letter]

6 Apr 2023

PONE-D-22-31465R1 

Leaf functional traits and pathogens: Linking coffee leaf rust with intraspecific trait variation in diversified agroecosystems 

Dear Dr. Isaac:

I'm pleased to inform you that your manuscript has been deemed suitable for publication in PLOS ONE. Congratulations! Your manuscript is now with our production department. 

Kind regards, 

on behalf of

Dr. Lukas Beule 

Academic Editor

PLOS ONE